# Study on the Farmland Improvement Effect of Drainage Measures under Film Mulch with Drip Irrigation in Saline–Alkali Land in Arid Areas

Li Zhao [1,2], Tong Heng [1,2,*], Lili Yang [1,2], Xuan Xu [1,2] and Yue Feng [1,2]

1   College of Water and Architectural Engineering, Shihezi University, Shihezi 832003, China; zhaoli_shzu@163.com (L.Z.); gemini-18@163.com (L.Y.); xuxuan319@163.com (X.X.); fengyue@stu.shzu.edu.cn (Y.F.)
2   Xinjiang Production and Construction Group Key Laboratory of Modern Water-Saving Irrigation, Shihezi 832003, China
*   Correspondence: htshz121@163.com

**Abstract:** Water scarcity and imbalances in irrigation and drainage are the main factors leading to soil salinization in arid areas. There is a recognized need for effective drainage measures to prevent and improve saline−alkali land. The principal objective of this project was to investigate the effects of drainage measures on soil desalination and farmland drainage in the process of improving saline–alkali soils; these measures included subsurface pipe drainage (SPD) and open ditch drainage (ODD). The results of the tests, conducted over two years, revealed that the soil desalination rate in the SPD test area was between 25.8% and 35.2%, the cotton emergence rate was 36.7%, and a 3.8 t hm$^{-2}$ seed cotton yield could be obtained. The soil electrolytic conductivity (EC) decreased step by step over time, and the average annual decrease reached 10 dS m$^{-1}$. The degree of soil salinization was reduced from a moderately saline soil level (8−15 dS m$^{-1}$) to a weakly saline soil level (4–8 dS m$^{-1}$). Thus, the phased goal of improving saline–alkali land was achieved. The soil desalination rate in the ODD test area was only 1/10 of the SPD area; high soil EC (9−12 dS m$^{-1}$) and groundwater level (2–3 m) were the most limiting factors affecting cotton growth in the ODD test area. The current results show that the critical depth of groundwater level affecting farmland secondary salinization is 4 m. In order to improve the salt discharge standard, SPD technology should be used on the basis of ODD. For salt that has accumulated in the soil for a long time, the technical mode of drip irrigation and leaching, followed by SPD drainage, in combination with the current irrigation system can achieve the goal of sustainable agriculture development.

**Keywords:** subsurface pipe drainage; open ditch drainage; soil desalination; saline−alkali land; drip irrigation; sustainable

## 1. Introduction

Soil salinization is one of the major factors limiting the sustainable development of agriculture in arid areas [1,2]. The long-term imbalance created by irrigation and leaching leads to large areas of soil secondary salinization [3,4]. Irrigation agriculture is key to increase agricultural productivity and ensure food security in arid areas [5,6]. Since the 21st century, drip irrigation systems have replaced the original canal irrigation systems, which makes it difficult to achieve irrigation and drainage balance [7,8]. Addressing the question of soil salinization under the current drip irrigation system will aid in further development of agricultural water-saving technology in arid areas [9]. The treatment of saline–alkali soil mainly lies in water conservancy measures [10], i.e., establishing a complete irrigation and drainage system and using leaching to remove excess salt in the soil. Farmland drainage measures are known to orchestrate the timely resolution of flooding and saline–alkali soil as well as improve soil texture and permeability; and they are an important tool to ensure

sustainable irrigation [11,12]. Wang [13] suggested that the prerequisite for maintaining the water–salt balance of farmland in arid areas is to strengthen the construction of open ditch drainage systems and at the same time to add a subsurface pipe drainage (SPD) system. Zhang et al. [8] summarized the current problems of farmland drainage and conducted field experiments; they believed that open ditch drainage (ODD) was suitable for areas with high groundwater levels downstream of the reservoir and that shaft drainage was suitable for areas with poor soil permeability in the plains. Previous studies have shown that cross-staggered ODD could improve soil density and porosity, increase soil temperature, and significantly increase field crop yield [14].

In arid areas, irrigation must not only support crop growth but also regulate the amount of water required due to soil salinity leaching [15,16]. In the past two decades, drip irrigation systems, which can increase soil moisture and prevent soil secondary salinization, have been widely implemented in cotton cultivation in Xinjiang to combat drought and water shortages. However, drip irrigation systems can only regulate the soil salinity in the root layer of crops, which cannot fundamentally decrease the soil salinity. In addition, drip irrigation systems have a certain uplifting effect on the groundwater level and pose a potential threat to the growth of crops in areas with higher groundwater levels. Therefore, drip irrigation systems can be used to further improve farmland drainage systems, change the original flooding irrigation leaching mode, and improve drainage standards, in order to solve the above problems. Drainage systems such as subsurface pipes and open ditch drainage systems can effectively solve the problem of soil salinization and control the groundwater level. Drainage systems combined with drip irrigation can directly and effectively decrease soil salinity and groundwater levels in unsaturated root zones, thus achieving the purpose of improving saline cultivated land. Li et al. [17] indicated that the current drainage technology of countries around the world has gradually changed from above ground to underground. Sallam [18] pointed out that Egypt invests at least $7.5 \times 10^8$ euros per year to increase the drainage amount of $1.0–1.5 \times 10^5$ hm$^2$. If SPD is not provided and soil salinization occurs, crop yield will decrease by 20% [19]. Chen [20] compared ODD and SPD from the aspects of project construction cost, cycle and maintenance management, and technical characteristics, and found that SPD has outstanding advantages compared with ODD. However, in actual agricultural production, there is no systematic experimental study on the effects of different drainage measures on the improvement of saline–alkaline soil and whether drainage measures have an impact on groundwater and crops.

This research study fills a gap in the literature by examining drainage measures that improve saline–alkali land in arid areas and using the geographic advantages of drip irrigation in the saline–alkali areas in the desert oasis in Xinjiang, China. SPD and ODD areas were selected for experimental monitoring over two years. We investigated the effects of soil desalination and farmland drainage under two drainage measures and analyzed the impact of the drainage system on groundwater and cotton growth in order to provide a theoretical foundation for the improvement of saline–alkali land in arid water-saving irrigation areas.

## 2. Materials and Methods

### 2.1. Site Description

The selection of the test area was based on the following principles: (a) geographic location in arid inland saline–alkali areas, (b) farmland with drainage measures, drip irrigation, and film-mulched land, (c) similar soil types and hydrogeological conditions. Based on this, two abandoned saline–alkali fields were selected in the north of AnJiHai Township, Shawan County, Xinjiang, China; these areas included subsurface pipe drainage (SPD) and open ditch drainage (ODD). The SPD (85°21′ E, 44°36′ N) and ODD test areas (85°39′ E, 44°56′ N) were 5.0 km apart, with planting areas of 3.4 hm$^2$ and 12 hm$^2$, respectively (Figure 1). This region occupies a mid-Atlantic zone with an extreme arid continental desert climate; it has an average annual sunshine of 2447.9 h. Throughout the

entire monitoring period, the average minimum and maximum temperatures were $-19.4$ and $31.4\,^{\circ}$C, respectively. The snowfall period in winter was from November to March each year. The mean annual precipitation was only 182.5 mm, while annual evaporation reached 1720 mm. The groundwater depth was 2–3 m during the non-irrigation period. The main soil textures were sandy loam and silt loam, and soil pH ranged from 7.51 to 8.53. The soil EC of the 0–20 and 20–40 cm soil layers at the beginning of the study was 14.3–16.1 and 12.9–15.5 dS m$^{-1}$, respectively. The main soluble salts in the soil were sulphate and chloride.

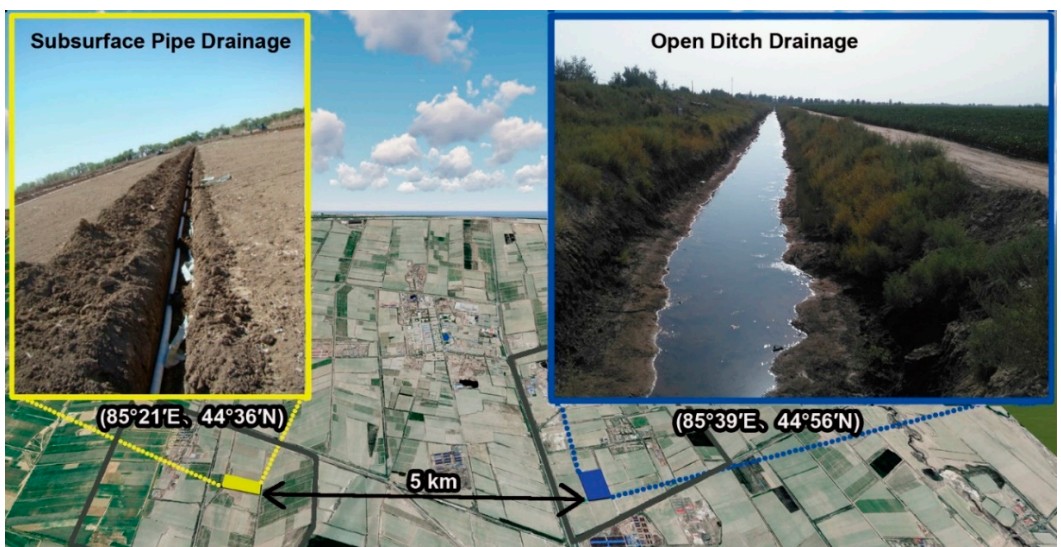

**Figure 1.** The locality map of subsurface pipe drainage (SPD) and open ditch drainage (ODD).

### 2.1.1. Open Ditch Drainage (ODD)

The terrain of the ODD test area was high in the northeast and low in the southwest, with a natural slope of 0.18% from the north to the south. The west side was close to an open ditch with a depth of 5.0 m. This is the most important drainage ditch connecting the Anjihai (AID) and Xiayedi (XID) irrigation districts [21].

### 2.1.2. Subsurface Pipe Drainage (SPD)

Terrain of the SPD test area was high in the southeast and low in the northwest, with a natural slope of 0.24% from the north to the south. This farmland was reclaimed in 1997, and cotton was planted under film mulch with drip irrigation. Affected by the accumulation of soil salinization, the cotton harvest decreased year over year from 2010 to 2014, and the area became abandoned farmland in 2015. In February 2016, the field was surveyed before the construction of the subsurface pipe; this survey included the meteorology, hydrogeology, soil and vegetation composition, field rodent path, blind ditch, etc. The main design parameters of SPD include slope, pipe diameter, buried depth, and spacing. The pipe diameter is determined according to the design drainage flow ($Q_1$, Formula (1)).

$$\begin{cases} Q_1 = CqA \\ d_1 = 2\left(\dfrac{nQ}{\alpha\sqrt{3i}}\right)^{3/8} \\ d_2 = 2\left(\dfrac{nQ}{\alpha\sqrt{i}}\right)^{3/8} \end{cases} \tag{1}$$

where $Q_1$ is the design drainage flow, m$^3$ d$^{-1}$; $C$ is the drainage flow reduction factor, when the drainage control area is less than 16 hm$^2$, $C = 1$; $A$ is the drainage control area, m$^2$; $d_1$ and $d_2$ are the diameter of subsurface pipe and water collecting pipe, mm; $i$ is the design slope, % (the slope should meet the requirement that the drainage flow is not less than 0.3 m s$^{-1}$; when the pipe diameter is less than 100 mm, the slope is 0.1–0.4%, and when the

pipe diameter is greater than 100 mm, the slope is 0.17–0.06%); $n$ is the roughness in the pipe, $n = 0.016$; $q$ is the designed drainage modulus to prevent salinization, m d$^{-1}$; $\alpha$ is the coefficient related to the filling degree, and the calculation formula of $\alpha$ is as follows:

$$\begin{cases} q = \dfrac{\mu\Omega(h_t - h_0)}{t} - \overline{\varepsilon_h} \\ \alpha = \dfrac{\left[\pi - \frac{\pi}{180}cos^{-1}(2\theta - 1) + 2\theta(2\theta - 1)\sqrt{\frac{1}{\theta}}\right]^{5/3}}{\left[2\pi - \frac{\pi}{180}cos^{-1}(2\theta - 1)\right]^{2/3}} \end{cases} \tag{2}$$

where $\overline{\varepsilon_h}$ is the average groundwater evaporation intensity, m d$^{-1}$; $\mu$ is the average water yield within the range of groundwater drawdown; $\Omega$ is the groundwater surface shape correction coefficient in the buried pipe area, $\Omega = 0.8 - 0.9$; $h_t$ and $h_0$ are threshold groundwater horizon and initial groundwater level, m; $\theta$ is the filling degree in pipe—when the pipe diameter is less than 100 mm, the filling degree is 0.7; $t$ is the leaching time, h. According to Formula (1), Formula (2), and the survey situation in SPD test area, the subsurface pipe slope and diameter are 0.4% and 90 mm, respectively.

The spacing and depth of the subsurface pipe were designed using the water balance principle and Hooghoudt's equation [22–24], as shown in Formula (3):

$$\begin{cases} H = h_k + \Delta h + d \\ L = \sqrt{\dfrac{4K_a h_t^2}{q} + \dfrac{8K_b e h_t}{q}} \end{cases} \tag{3}$$

where $H$ is the buried pipe depth, m; $h_k$ is the critical depth of groundwater, depth of drainage, or depth of soil improvement, m; $\Delta h$ is the retained head, m; $d$ represents the pipe diameter, m; $h_t$ represents the head of water midway between drains, m; $K_a$ and $K_b$ are the soil hydraulic conductivity at above-drain and below-drain levels, m day$^{-1}$; $q$ is the design drainage rate, m day$^{-1}$; $L$ is the drain spacing, m; $e$ is the Hooghoudt's equivalent depth, m. The experimental design subsurface pipe spacing was 15 m, and the laying depth and length of each subsurface pipe were 1.0 m.

Construction of the SPD test area began in early March 2016 and finished at the end of April 2016. Details of the experimental installation process are as follows. A soil profile was excavated using a hydraulic excavator (Doosan 331), and the subsurface pipes were then placed horizontally in the soil profile. Sand and gravel filter material (with particle sizes ≤4 cm) was backfilled around the pipes to a thickness of 20 cm. The subsurface pipes were backfifilled with soil layer by layer to complete the construction. High-quality resin integrated water collection wells were set in the end of each drainage pipe, and the wells were linked by connection pipes, which ultimately discharged to open drainage ditches. All the backfilled soil layers were compacted layer by layer, except those that were within 20 cm of the filter material.

### 2.1.3. Crop Management—Drip Irrigation and Leaching

The measurement of leaching and drainage is key to the improvement of saline–alkali land in arid areas. We conducted three leaching tests, and the selected leaching dates were 8 June 2016 (DL1), 8 September 2016 (DL2), and 18 April 2017 (DL3). Leaching water was from snowmelt in the Tianshan Mountains (the salinity of which was 0.8 g L$^{-1}$). Leaching water demand was determined by soil EC and the critical salinity of the allowed crop growth [25].

$$D_W / D_s = -C\lg[(EC_a - 2EC_i)/(EC_s - 2EC_i)] \tag{4}$$

where $D_W$ (m) is leaching water volume, $D_S$ (m) is leaching need of soil layer depth, $EC_a$ (dS m$^{-1}$) is the critical salinity of the allowed crop growth, $EC_i$ (dS m$^{-1}$) is the irrigation water salinity, $EC_s$ (dS m$^{-1}$) is initial soil EC, and $C$ is the salt leaching coefficient ($C = 1.06$).

Early in the trial, the salt content of the shallow soil layers (0–40 cm) was relatively high. We consider that oil sunflower (*Helianthus annuus* Linn.) has higher salt tolerance than cotton (*Gossypium hirsutum* L.), and oil sunflower can be used as a green manure crop

to effectively improve soil quality. Therefore, oil sunflowers were sown on 8 June 2016 using cultivar KF366 (KeFeng®, China). The growth period of oil sunflower was about 3 months, which could be subdivided into four growth periods: seedling, bud, flowering, and maturity stages (Table 1). The oil sunflowers were cultivated in wide/narrow planting rows, with a wide-row spacing of 60 cm and narrow-row spacing of 30 cm. The transparent plastic film mulch width was 140 cm, and the single-hole flow of drip irrigation tape was 2.6 L h$^{-1}$, with a dripper spacing of 30 cm and operating pressure of 0.09 MPa. A total of 350 kg ha$^{-1}$ ($CH_4N_2O$) urea was applied with irrigation water throughout the oil sunflower growth season. The amount of compound fertilizer used was 180 kg ha$^{-1}$ (15% P2O5: 10% K2O: 9% ZnSO4·7(H2O): 9% H3BO3), and in total the area was irrigated 8 times.

**Table 1.** Irrigation and leaching schedule during the oil sunflower (2016) and cotton (2017) growing season.

| Cotton Growth | Irrigation Date | Irrigation Time (h) | I Quota (m$^3$ ha$^{-1}$) | Oil Sunflower Growth | Irrigation Date | Irrigation Time (h) | I Quota (m$^3$ ha$^{-1}$) |
|---|---|---|---|---|---|---|---|
| Squaring | 20 May | 30 | 450 | Seedling | 8 Jun | 33 | 800 |
| | 10 Jun | 24 | 600 | | | | |
| | 25 Jun | 25 | 600 | | 25 Jun | 30 | 675 |
| Flowering | 5 Jul | 25 | 600 | Squaring | 5 Jul | 31 | 675 |
| | 11 Jul | 24 | 600 | | 16 Jul | 30 | 675 |
| | 20 Jul | 24 | 600 | | 25 Jul | 30 | 675 |
| Bolling | 5 Aug | 25 | 600 | Flowering | 8 Aug | 31 | 675 |
| | 10 Aug | 24 | 600 | | 13 Aug | 30 | 675 |
| Boll opening | 19 Aug | 25 | 600 | Maturity | 26 Aug | 31 | 675 |
| Leaching Scheme | 1st Drip Irrigation Leaching (DL1) | | | 2nd Drip Irrigation Leaching (DL2) | | 3rd Drip Irrigation Leaching (DL3) | |
| Leaching date | 8 Jun 2016 | | | 8 Sep 2016 | | 18 Apr 2017 | |
| Total quota (m$^3$ ha$^{-1}$) | 8000 | | | 7500 | | 7500 | |
| Leaching time (h) | 60 | | | 64 | | 56 | |

Cotton was sown on 20 April 2017 using cultivar XLZ. 66 (Huiyuan®, China). The growth period of cotton was about 4 months, which could be subdivided into four growth periods: seedling, squaring, flowering, and bolling stages. Cotton sowing involved the application of film mulch with drip irrigation technology; the transparent plastic film mulch width was 205 cm. Three drip irrigation tapes and six cotton planting rows were used. The spacing between drip irrigation tapes was 75 cm, the single-hole flow of the drip irrigation tape was 2.6 L h$^{-1}$, and the cotton planting rows had a wide-row spacing of 45 cm and narrow-row spacing of 30 cm. A total of 260 kg ha$^{-1}$ ($CH_4N_2O$) urea was applied with irrigation water throughout the cotton growth season. The amount of compound fertilizer used was 200 kg ha$^{-1}$ (15% P2O5: 10% K2O: 9% ZnSO4·7(H2O): 9% H3BO3). In addition, the nitrogen input by the foliar fertilizer during pesticide spraying was about 10−20 kg ha$^{-1}$. In total, the area was irrigated 9 times.

### 2.2. Experiment Design

In the SPD test area, 0.5 m ($P_1$), 5 m ($P_2$), and 7.5 m ($P_3$) horizontal distance from the subsurface pipe were set as the first factor. In the ODD test area, 0.5 m ($D_1$), 30 m ($D_2$), and 60 m ($D_3$) horizontal distance from the open ditch were set as the second factor. Thus, there was a total of 6 treatments (Figure 2).

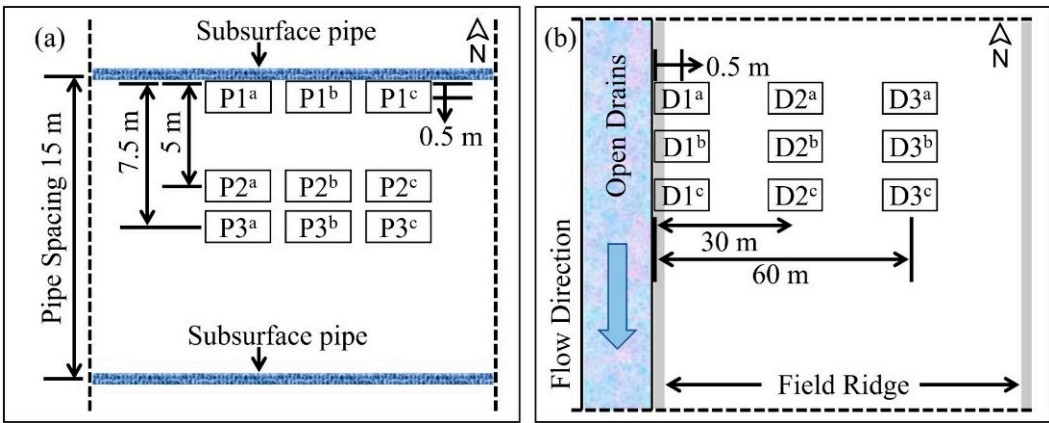

**Figure 2.** Study planning diagrams of subsurface pipe drainage (**a**) and open ditch drainage (**b**).

*2.3. Sampling, Measurements, and Calculations*

2.3.1. Soil Electrical Conductivity (EC) and Desalination Rate

Soil samples were collected in the middle of each month. A soil auger with a 5 cm inside diameter was used to collect 0–20 and 20–40 cm depth soil samples, and random sampling was repeated 3 times. All soil samples were dried with a DHG-9003 drying at 150 °C (Shanghai Leichi®, China). Once dried, 10 g of each sample was removed, ground, sifted through a 1 mm sieve, and then placed in a 150 mL Erlenmeyer flask with 50 mL of distilled water. We used an Oscillatorsto to stir the Erlenmeyer flask (water-to-soil mass ratio was 1:5) for 10 min (Shanzhi®, China). After standing for 15 min, soil EC of the water extract was measured using a DDS-11 conductivity meter (INESA®, China). Soil desalination rate was calculated using Formula (5).

$$D_r = [(EC_1 - EC_2)/EC_1] \times 100\% \tag{5}$$

where $D_r$ is the desalination rate, %; $EC_1$ is the initial value of soil salinity, dS m$^{-1}$; $EC_1$ is the fifinal value of soil salinity after irrigation, dS m$^{-1}$. In this study, the whole test period was divided into 4 experimental stages when calculating the soil desalination rate: L1 (2 March 2016 to 18 June 2016), L2 (18 June 2016 to 8 September 2016), L3 (8 September 2016 to 23 April 2017), and L4 (23 April 2017 to 24 December 2017).

2.3.2. Drainage Flow and Drainage Water Salt Concentration

The drainage flow and drainage water salt concentration of SPD and ODD were monitored during farmland leaching and drainage. The monitoring started at the beginning of the drainage stage of subsurface pipes and was performed every 2–6 h. Because the space in the water collection well was limited, the following method was used to measure the drainage flow accurately. First, the water from the well was collected in a water tank. After 10 s, the water tank was lifted from the well and the water volume was measured using graduated cylinders when the water depth became static. The drainage flow measurement was repeated four times. The tank capacity was 10 L, and the graduated cylinders were 500, 1000 and 2000 mL. Finally, the collected water samples were brought back to the laboratory in a cooler and stored at 4 °C for drainage water salt concentration analysis within 48 h.

In the ODD test area, drainage flow was monitored by a pipeline flowmeter (LDG−MIK, Hangzhou Meacon®, Hangzhou, China), and the sampling time was the same as for the SPD test area.

2.3.3. Groundwater Level and Mineralization

The groundwater observation wells were set up in the SPD and ODD test areas to monitor the shallow groundwater level and mineralization, and the monitoring date was in the middle of every month (March 2016–November 2017). The collected groundwater

samples were taken back to the laboratory, and the mineralization was determined by the oven-drying method.

### 2.3.4. Crop Growth and Yield

Oil sunflower seedling emergence rate, plant height, and dry matter mass were monitored during the growth period in 2016. A rectangular area of $2 \times 2$ m (with three repeats) was selected randomly within the experimental area to measure the seeding emergence rate every 7 d after irrigation during the seedling period. We selected three representative oil sunflowers during each growth period and measured plant height from the main stem to the top of the main stem 2 weeks after sowing. After the plant height measurement was completed, the roots, stems, leaves and fruit organs of the oil sunflower were collected and dried at 105 °C for 24 h. Once dried, we determined the dry matter mass. These plants were observed once a week.

The detection methods of cotton seedling emergence rate, plant height, and dry matter quality were consistent with those used for oil sunflowers. When cotton entered the full boll-opening (BOf), the seed cotton yield was determined by hand harvesting in multiples of three.

## 3. Results

### 3.1. The Spatiotemporal Patterns of Soil EC

In arid areas, the primary function of the farmland drainage system is to discharge excess water in the soil. Drainage measures play a role in reducing both the groundwater level and soil salinity. The soil EC in the 0–20 cm and 20–40 cm soil layers in the SPD and ODD test areas showed a significant difference ($p < 0.05$) in change over time (Figure 3). Between March 2016 and September 2017, the soil EC in the SPD test area showed a stepped decrease over time, with this trend clustered mostly in the period of drip irrigation and leaching; the specific performance was P1 < P2 < P3. These data indicate that the closer the horizontal distance to the subsurface pipe, the smaller the soil EC. The soil EC in the ODD test area showed a decrease slowly over time; the specific performance was D3 < D2 < D1. The closer the horizontal distance to the open ditch, the larger the soil EC. This result was diametrically opposite to the soil EC in the SPD test area. A reasonable explanation was the SPD test area was only drained during the period of drip irrigation and leaching, and the farmland had no drainage measures the rest of the time. When it was not drained in water, the soil salt was mainly transferred to the surface soil by transpiration. The ODD test area was continuously drained, and the salt in the soil continuously migrated to the open ditch. Therefore, the overall performance of soil EC was P1 < P2 < P3 < D3 < D2 < D1. After each drip irrigation and leaching (DL1, DL2, and DL3) in the SPD test area, soil EC showed an increasing trend, and this phenomenon appeared in each treatment in 0–20 and 20–40 cm soil layers. Among them, the P1 treatment had the largest increase of soil EC after DL2 ended, reaching 1.2 dS m$^{-1}$. The soil EC showed a slow decreasing trend after drip irrigation and leaching in the ODD test area. In addition, the decreasing trend of soil EC in the ODD test area was less than the increasing trend of SPD. Among them, D1 and D2 treatments had the largest decrease of soil EC after DL3, which was 1.0 dS m$^{-1}$. Obviously, in the SPD test area, the decrease of soil EC in the 20–40 cm soil layer was higher than in the 0−20 cm soil layer, and the difference of soil EC in the 20–40 cm soil layer was significantly less than in the 0−20 cm soil layer.

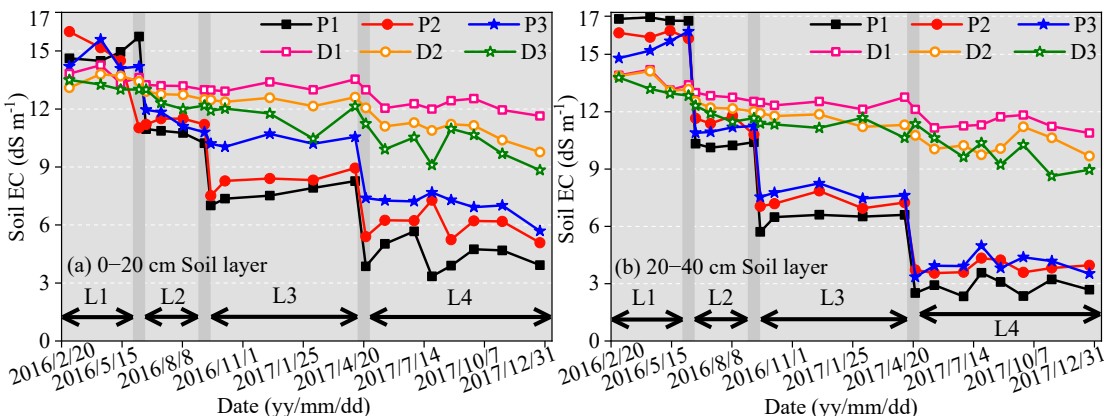

**Figure 3.** Temporal evolution of soil EC in 0–20 (**a**) and 20–40 (**b**) cm soil layer from 2016 to 2017.

Throughout the experimental period, the soil EC of P1, P2, P3, D1, D2, and D3 treatments decreased by 10.7, 9.9, 8.5, 2.2, 3.3, and 4.7 dS m$^{-1}$ in the 0–20 cm soil layer, respectively; the soil EC of P1, P2, P3, D1, D2, and D3 treatments decreased by 14.7, 12.2, 11.3, 3.0, 4.2, and 4.8 dS m$^{-1}$ in the 20−40 cm soil layer, respectively. There was no significant decrease of soil EC in 0–20 and 20–40 cm soil layers.

In general, in the SPD test area, the decrease of soil EC was the smallest in the middle position of the two concealed pipes (P3) and the largest in the horizontal distance of 0.5 m from the concealed pipes (P1). The 20–40 cm soil layer saw the largest decrease; after the experiment, the soil EC of P1, P2, and P3 treatments was lower than 5.0 dS m$^{-1}$. This implies that the average annual decrease of the soil EC in the 0–40 cm soil depths in the SPD test area reached 10 dS m$^{-1}$. In the ODD test area, the further away from the open ditch, the more the soil EC decreased; however, the soil EC was still between 9–12 dS m$^{-1}$.

### 3.2. Effectiveness of Soil Desalination

Table 2 shows the results of the soil desalination analysis, where the leaching was performed once in the LI, L2, and L3 stages, i.e., DL1, DL2, and DL3. Throughout the experimental period, the average desalination rate of the P1, P2, P3, D1, D2, and D3 treatments was 35.2, 27.5, 25.8, 2.8, 5.3, and 6.9%, respectively. Among them, the soil desalination rate of the P2 treatment in the L4 stage as well as the D1, and D3 treatments in the L3 stage presented negative values and were predominantly localized in the 20–40 cm soil layer; thus, the above three treatments showed a trend of salt accumulation in the 20–40 cm soil layer in later trials. In the L1, L2 and L3 stages, the soil desalination rate of the P3 treatment in the 0−20 cm soil layer was significantly lower than the P1 and P2 treatments ($p < 0.05$) in the SPD test area, and the P1 treatment in the 20–40 cm soil layer was significantly higher than the P2 and P3 treatments ($p < 0.05$). The differences were not significant in any case in the ODD test area ($p > 0.05$). In the L4 stage, the soil desalination rate of the P1, P2, and P3 treatments in the 0–20 cm soil layer was not significant ($p = 0.185$), but the P2 treatment in the 20−40 cm soil layer was significantly lower than the P1 and P3 treatments ($p = 0.015$). The soil desalination rate of the D1 treatment in the 0−20 cm soil layer was significantly lower than the D2 and D3 treatments ($p = 0.036$) in the ODD test area, and the D3 treatment in the 0–20 cm soil layer was significantly higher than the D1 and D2 treatments ($p < 0.035$). In addition, we also found that the soil desalination rate of the SPD and ODD areas had significant similarities and differences ($p < 0.05$) in the 0–20 cm soil layer during the L1, L2 and L3 stages and that the difference in the SPD and ODD areas in the 20–40 cm soil layer during the L4 stage was significant. All the other differences were non-significant.

**Table 2.** Soil desalination rate of SPD and ODD, 0.5 (P1), 5 (P2), and 7.5 m (P3) horizontal distance from the subsurface pipe, 0.5 (D1), 30 (D2), and 60 m (D3) horizontal distance from the open ditch for L1 (2 March 2016 to 18 June 2016), L2 (18 June 2016 to 8 September 2016), L3 (8 September 2016 to 23 April 2017), and L4 (23 April, 2017 to 24 December 2017).

| Treatment | 0–20 cm Soil Layer (%) | | | | 20–40 cm Soil Layer (%) | | | | AVG [1] (%) |
|---|---|---|---|---|---|---|---|---|---|
| | L1 | L2 | L3 | L4 | L1 | L2 | L3 | L4 | |
| Subsurface pipe drainage | | | | | | | | | |
| P1 | 25.0 a | 35.5 a | 47.6 a | 21.9 a | 38.8 a | 43.6 a | 61.3 a | 8.2 a | 35.2 a |
| P2 | 30.0 a | 34.7 a | 34.9 b | 18.6 a | 27.7 b | 35.0 b | 48.2 b | −11.5 c | 27.6 b |
| P3 | 15.8 b | 17.9 b | 26.4 c | 21.6 a | 26.4 b | 31.1 b | 56.9 b | 10.6 a | 25.8 b |
| Open ditch drainage | | | | | | | | | |
| D1 | 4.2 c | 1.7 c | −0.5 d | 3.4 c | 6.5 c | 2.8 c | 1.7 c | 2.4 b | 2.8 c |
| D2 | 1.5 c | 2.3 c | 2.5 d | 10.9 b | 9.5 c | 2.4 c | 8.6 c | 3.7 b | 5.3 c |
| D3 | 3.6 c | 3.2 c | 6.6 d | 12.0 b | 10.7 c | 4.4 c | −0.2 c | 15.6 a | 6.9 c |
| ANOVA (*p*−value) | | | | | | | | | |
| $P_a$ (df1) | 0.041 * | 0.024 * | 0.005 ** | 0.185 | 0.047 * | 0.038 * | 0.029 * | 0.015 * | 0.022 * |
| $D_a$ (df2) | 0.075 | 0.083 | 0.067 | 0.036 * | 0.121 | 0.156 | 0.087 | 0.035 * | 0.088 |
| $P_a \times D_a$ (df2) | 0.021 * | 0.064 * | 0.034 * | 0.069 | 0.284 | 0.076 | 0.097 | 0.041 * | 0.068 |

[1] AVG is the average value, ** significant at 0.01, * significant at 0.05, different letters (a, b, c) represent a significant difference within groups.

In general, the soil desalination rate in the SPD test area ranged between 25.8–35.2%; in the ODD test area, it ranged between 2.8–6.9%. This suggests that the soil desalination efficiency of open ditch drainage (ODD) was only 1/10 of subsurface pipe drainage (SPD), especially during the third drip irrigation and leaching (DL3), and that the differences between P1, P2, and P3 treatments were extremely significant (*p* = 0.005). Even if the amount of drip irrigation water was increased, the salt removal efficiency of ODD was much lower than in the SPD test area.

The significance of different treatments was statistically tested, and data were tested for normality (Kolmogorov-Smirnov test) and homogeneity of variances (Levene's test). The application of the Kolmogorov-Smirnov test showed that the baseline data are non–homogeneous (*p* = 0.004) and that the soil desalination rate of SPD and ODD test areas are not drawn from the same distribution; thus the differences between these distributions are statistically significant (*p* < 0.05). Alternatively, the Q-Q plots showed that the soil desalination rate of each treatment did not follow the normal distribution (Figure 4). The Levene statistic value was 9.778. Both Welch and Brown–Forsythe tests of equality of means were also significant: the Welch statistic value was 525.1 and the Brown–Forsythe statistic value was 117.11 (*p* < 0.01). For the one-way ANOVA test, F value was 129.9, which was smaller than $F_{0.01}$. Consequently, the difference intergroup was statistically significant.

The soil desalination rate of the SPD and ODD test areas was analyzed using the Tukey multiple comparison test, and the results all showed statistical significance (*p* = 0.001). The analysis results corroborated the previous expectation, i.e., the effectiveness of soil desalination using subsurface pipe drainage (SPD) was significantly better than open ditch drainage (ODD) (Table 3).

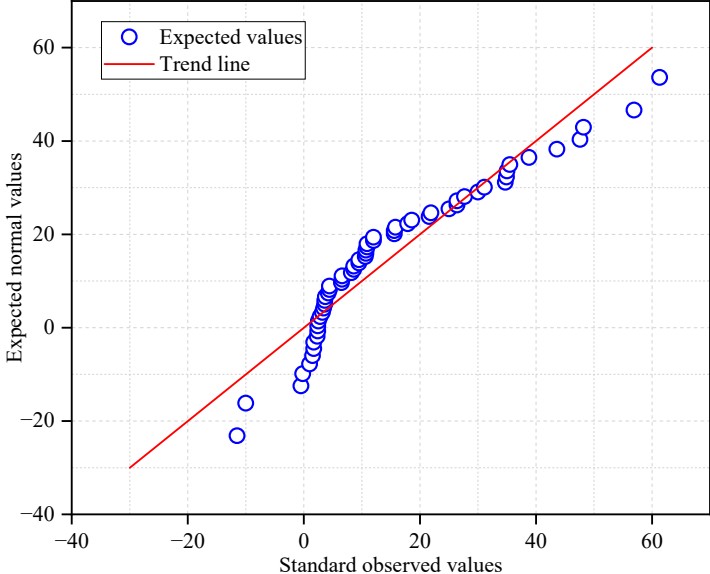

**Figure 4.** Normal Q-Q plot of soil desalination rate.

**Table 3.** Test of difference between SPD and ODD.

| Difference Source | Intergroup | Intragroup | Total | Multiple Comparisons | |
|---|---|---|---|---|---|
| Sum of squares | 6682.157 | 257.084 | 6939.241 | MD [3] | 51.83167 * |
| df [1] | 6 | 8 | 48 | SEM [4] | 3.29607 |
| RMS [2] | 3341.079 | 25.708 | | *p*-value | 0.001 |
| F-value | 129.961 | | | 95% CI [5] | 44.4876 |
| F-test | F < F$_{0.01}$ | | | | 59.1758 |
| *p*-value | 0.004 | | | | |

[1] df is the degree of freedom, [2] RMS is the root mean square, [3] MD is the mean difference, [4] SEM is the standard error, [5] CI is the confidence interval, * significant at 0.05.

### 3.3. Dynamic Variation of Drainage Flow and Salt Concentration of the Drainage Water

The spatiotemporal patterns of drainage measures under film mulch with drip irrigation and leaching were analyzed in the previous section; they indicated that soil EC decreased in 0–40 cm depth soil. However, whether soil EC was decreased by SPD or ODD, or by deep leakage, shallow evaporation, etc., needs to be confirmed by the dynamic variation of leaching test.

Through the monitoring of the drainage flow and salt concentration of the drainage water (Figure 5), we found that the drainage duration of DL1, DL2, and DL3 was 90, 84, and 90 h, respectively. The drainage flow in the SPD test area showed a similar pattern and presented a unimodal pattern. In contrast, the drainage flow exhibited a slight change from 3.0 to 5.0 m$^3$ h$^{-1}$ in the ODD test area. The maximum drainage flows of the SPD and ODD test areas during DL1, DL2, and DL3 were 2.0 and 4.0, 3.2 and 3.7, 2.2 and 4.9 m$^3$ h$^{-1}$, respectively, and they occurred at 35 and 40, 32 and 50, 38 and 26 h, respectively. The salt concentration of the drainage water was used to represent the total amount of inorganic mineral components in the water (or total salt content). We found that the salt concentration of the drainage water in the SPD test area gradually decreased as the leaching times (DL1 to DL3) increased, whereas the ODD test area showed an opposite trend. The maximum salt concentrations of the drainage water in the SPD and ODD test areas during DL1, DL2, and DL3 were 185 and 124, 158 and 136, 130 and 147 g L$^{-1}$, respectively, and they occurred at 50 and 68, 29 and 55, 43 and 37 h, respectively. After all three leaching events, the average salt concentration of the drainage water in the SPD test area was 180 g L$^{-1}$ to 120 g L$^{-1}$, and the decline was approximately 33%, and SPD test area increased from 120 g L$^{-1}$ to 140 g L$^{-1}$, and the rise was approximately 17%. In general, the drainage flow in the ODD

test area was higher than the SPD test area, and the drainage flow and salt concentration of the drainage water in the ODD test area had fluctuations of small magnitude, whereas SPD presented a unimodal pattern.

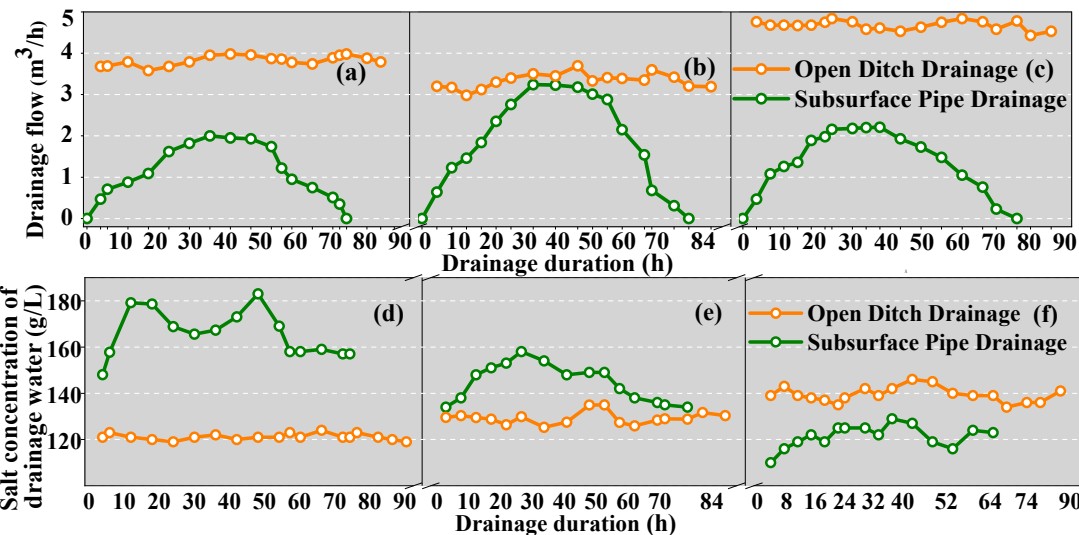

**Figure 5.** Drainage flow and the salt concentration of the drainage water: (**a**) DL1 drainage flow; (**b**) DL2 drainage flow; (**c**) DL3 drainage flow; (**d**) DL1 salt concentration of drainage water; (**e**) DL2 salt concentration of drainage water; (**f**) DL3 salt concentration of drainage water.

### 3.4. Dynamic Variation of Groundwater Level and Salt Concentration of Groundwater

Close relations were elicited between the cause of soil salinization and groundwater level uplift. Generally, the soil moisture carries salt seep between two stable waterproof layers during the flood season; during the dry and snowmelt season, the soil salt rises to the shallow soil layer through transpiration. Liu et al. [26] argued that the fundamental measures to improve soil salinization are to adopt farmland drainage measures, which can effectively control the groundwater level. To this end, we contrasted the groundwater level and salt concentration of groundwater in the SPD and ODD test areas from 2016 to 2017 (Figure 6). The groundwater level exhibited a slight change from 1.2 to 3.3 m in the ODD test area, and the highest groundwater level was 1.2 m, from April to May per year, when the local area had just passed the snowmelt period. The groundwater level in the SPD test area ranged from 1.5−5.8 m and fluctuated greatly; in the DL1, DL2, and DL3 stages, it rose by 1.6, 1.9, and 2.2 m, respectively. From the end of the third leaching to the last monitoring (28 April 2017 to 24 December 2017), the groundwater level in the SPD and ODD test areas showed decreasing trends to varying degrees, and the decline of groundwater levels was 2.8 and 1.1 m, respectively.

After the test trial, the groundwater level in the SPD test area declined below 5 m, and the risk of soil secondary salinization was relatively small, whereas the ODD test area was about 3 m, and there was still a risk of soil secondary salinization. In addition, the change trend of salt concentration of groundwater in the SPD test area had a strong linear correlation with the groundwater level ($R^2 = 0.83$), and the salt concentration of groundwater was from 40 to 110 g L$^{-1}$, with an average annual increase of 35 g L$^{-1}$ a$^{-1}$. The salt concentration of groundwater exhibited a slight change from 46 to 70 m$^3$ h$^{-1}$ in the ODD test area; the fluctuations were of less magnitude compared with SPD, with an average annual increase of only 12 g L$^{-1}$ a$^{-1}$.

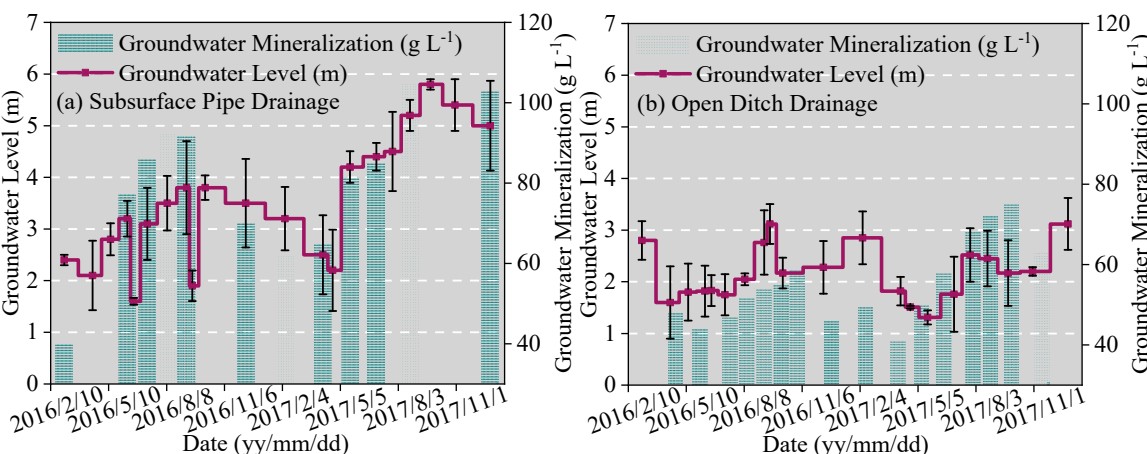

**Figure 6.** Dynamic variation of groundwater level and salt concentration of the groundwater in SPD (**a**) and ODD (**b**) test areas from 2016 to 2017. Dual y−axes represent groundwater level on the left and salt concentration of the groundwater on the right.

### 3.5. Crop Growth and Yield

The indicators of oil sunflower and cotton growth from 2016 to 2017 are shown in Table 4. During the 2016 oil sunflower planting, the emergence percentages of oil sunflower in the SPD and ODD test areas were above 45.6 and 39.5%, respectively, and D1, D2, and D3 treatments had no significant difference ($p = 0.076$). The oil sunflower height in the SPD and ODD test areas showed a similar pattern, i.e., the closer the horizontal distance to the subsurface pipe, the lower the plant height. The oil sunflower height of the P3 treatment was the highest, which was 110.5 cm. In the ODD test area, the differences between D1, D2, and D3 treatments were extremely significant ($p = 0.005$); the oil sunflower dry matter mass was not significantly different among the treatments, which was above 14.1 t ha$^{-1}$. During the 2017 cotton planting, the emergence percentage, plant height, dry matter mass, and seed cotton yield were not significantly different in the SPD and ODD test areas. Cotton emergence percentage and height were higher in P1 treatments; however, cotton dry weight was 14–12% less in the P1 treatment than P2 and P3 treatments, and seed cotton yield was 15% lower in T1 than T2 and T3. The results indicated that the SPD is favorable to cotton growth and seed yield compared to the ODD test areas. The seed cotton yield of the P2 and P3 treatments (3.8 t ha$^{-1}$) was higher than for the other treatments; among them, the seed cotton yield of the P3 treatment was 39–46% higher than ODD test area.

In general, the average seed cotton yield of the SPD and ODD test areas was 3.6 and 2.4 t ha$^{-1}$, respectively, but there was no significant difference between them ($p = 0.225$). The closer the horizontal distance to the subsurface pipe or open ditch, the lower the oil sunflower emergence percentage, plant height, dry matter mass, seed cotton yield, and cotton dry matter mass. The seed cotton yield had a stronger correlation with the cotton emergence percentage, plant height, and dry matter mass ($R^2 = 0.69-0.80$). However, high soil EC level (9–12 dS m$^{-1}$) and groundwater level (3 m) remained by far the most dominating limiting factors for plant growth in the ODD test area.

**Table 4.** The oil sunflower (2016) and cotton growth (2017), 0.5 (P1), 5 (P 2), and 7.5 m (P3) horizontal distance from the subsurface pipe, 0.5 (D1), 30 (D2), and 60 m (D3) horizontal distance from the open ditch.

| Treatment | Oil Sunflower (2016a) | | | Cotton (2017a) | | | |
|---|---|---|---|---|---|---|---|
| | EP [1] (%) | Height (cm) | DM [2] (t ha$^{-1}$) | EP [1] (%) | Height (cm) | DM [2] (t ha$^{-1}$) | SCY [3] (t ha$^{-1}$) |
| Subsurface Pipe Drainage | | | | | | | |
| P1 | 45.6b | 103.5b | 15.6a | 36.7a | 62.7a | 4.1b | 3.3b |
| P2 | 49.4a | 108.0a | 16.8a | 33.4a | 58.2a | 4.7a | 3.8a |
| P3 | 48.7a | 110.5a | 16.2a | 33.8a | 60.4a | 5.2a | 3.8a |
| Open Ditch Drainage | | | | | | | |
| D1 | 39.5b | 95.5c | 14.1a | 21.6b | 30.3c | 2.6c | 2.3c |
| D2 | 42.6b | 102.5b | 15.9a | 22.4b | 32.6c | 2.6c | 2.3c |
| D3 | 45.3b | 110.0a | 16.5a | 25.3b | 39.4b | 3.1c | 2.6c |
| ANOVA (*p*−value) | | | | | | | |
| P$_a$ (df1) | 0.043 * | 0.034 * | 0.120 | 0.185 | 0.095 | 0.042 * | 0.035 * |
| D$_a$ (df2) | 0.076 | 0.009 ** | 0.084 | 0.259 | 0.028 * | 0.162 | 0.295 |
| P$_a$ × D$_a$ (df2) | 0.062 | 0.027 * | 0.303 | 0.012 * | 0.035 * | 0.039 * | 0.225 |

[1] EP is the emergence percentage, [2] DM is the dry matter mass, [3] SCY is the seed cotton yield, ** significant at 0.01, * significant at 0.05.

Oil sunflower and cotton growth was in a transition stage when the soil EC was constantly decreasing (Figure 7). In 2016, the highest correlations were seen between the changes in soil EC and oil sunflower plant height, followed by emergence percentage (R2 = 0.64) and dry matter mass (R2 = 0.63). Likewise, in 2017, the highest correlations were seen between the change values in soil EC and cotton plant height (R2 = 0.86), followed by emergence percentage (R2 = 0.83), dry matter mass (R2 = 0.76), and seed cotton yield (R2 = 0.74). Overall, the oil sunflower plant height, emergence percentage, and dry matter mass status increased significantly when the soil EC change decreased from 9 to 5 dS m$^{-1}$; the cotton plant height, emergence percentage, dry matter mass, and seed cotton yield status increased significantly when the soil EC change decreased from 4 to 1 dS m$^{-1}$.

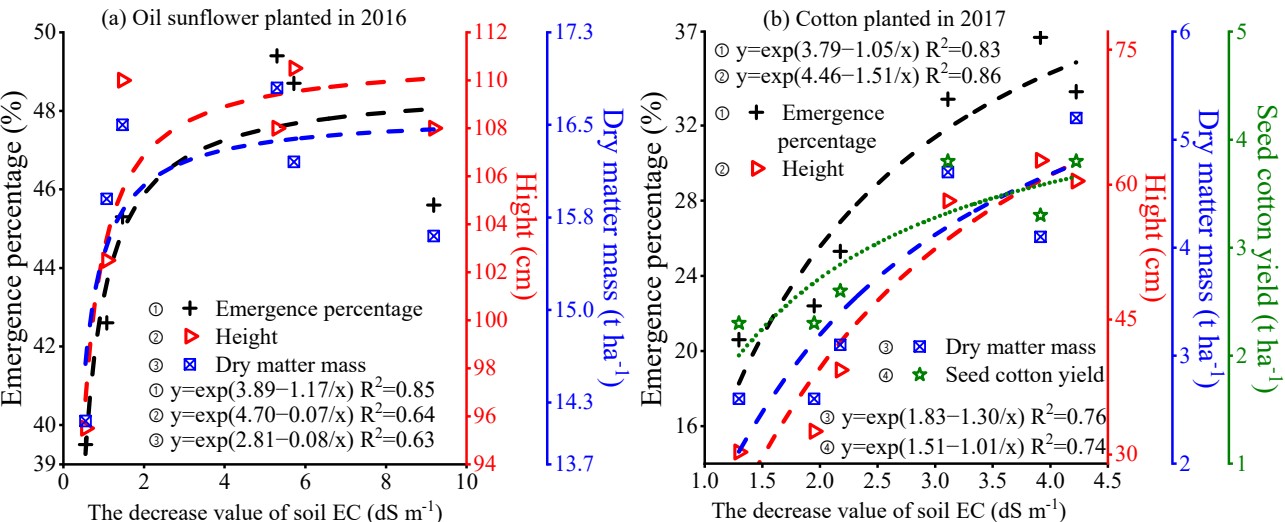

**Figure 7.** Effects of the soil EC decrease value on growth of oil sunflower (**a**) and cotton (**b**). Δ represents the threshold of soil EC.

## 4. Discussion

### 4.1. The Relationship Between Soil EC, Groundwater Level, and Salt Concentration of Groundwater

In arid and saline–alkali areas, the groundwater level of farmland is usually between 1–3 m. It is well known that saline-alkali soils are hardened and impervious. The traditional improvement method is to reduce the soil salt in the cultivated layer through ground leaching [27] so that the salt leaks to deeper soil layers; however, this raises the groundwater level and threatens the development of plant roots. Our approach was to reduce cultivated layer soil EC through SPD and ODD, with the initial purpose of decreasing groundwater levels when waterlogging occurs in farmland. However, if the groundwater level is lower than the depth of SPD or ODD, they cannot play a role in draining saline water. Wu et al. [28] studied the effect of SPD and ODD on reducing waterlogging and wheat growth, and the results showed that heavy rainfall caused the groundwater level of SPD to be 0.2 cm lower than ODD. However, the results of the present study showed an opposite trend, After the test, the groundwater level of the SPD test area was 2.1 m lower than for ODD. We combined drip irrigation and leaching with the drainage measures to decrease the shallow soil EC while controlling the groundwater level in order to meet the requirements of the crop being grown. Our study showed that the groundwater level decreased 1 m for every 3.3 dS m$^{-1}$ soil EC decrease in the SPD test area. However, the decrease of soil EC was accompanied by an increase in the salt concentration of groundwater of 63 g L$^{-1}$. In other words, the salt concentration of groundwater increased 21 g L$^{-1}$ for every 1.0 dS m$^{-1}$ soil EC decrease in 0–40 cm soil layers. In the one to four months after leaching and SPD measures, we found that the salinity of groundwater continued to increase, and the groundwater level also rose gradually; however, the upward trend lasted for up to four months and then began to gradually decrease (Figure 6). When leaching and SPD measures decreased the soil EC to a slightly saline level (1–3 dS m$^{-1}$), we reduced the frequency of SPD. In addition, our research also found that the high salt concentration of groundwater did not cause the rebound of soil EC in 0–40 cm soil layers in the SPD test area. Therefore, we speculated that the critical depth of groundwater level affecting soil secondary salinization of saline–alkali cultivated land was 4 m.

### 4.2. The Relationship between Soil Desalination Rate, Drainage Flow, and Salt Concentration of the Drainage Water

Drainage flow and salt concentration of the drainage water are the group of indicators that most directly measure soil desalination rate in farmland drainage measures [29]. Our results show that the desalination effect was higher in the high salinity environment during the initial stage of leaching, and it decreased as the soil EC decreased during the later stages of the leaching. At this point, a greater leaching water volume (DW) was needed to substitute the same amount of soil EC. The wetting front was constantly overlapping around the dripper. Meanwhile, the soil EC moved to drainage pipes under the hydrodynamic drive of drip irrigation. Moreover, drip irrigation leaching consumed less water than flood leaching over time. Hanson et al. [30] demonstrated that drip irrigation leaching that partially wets the soil surface area is highly efficient only under conditions of severe irrigation deficit, and the leaching fraction decreased over time. After all three leaching events, the average soil desalination rates were 35.2% (DL1), 27.6% (DL2), and 25.8% (DL3), and the average drainage flow and salt concentration of the drainage water were 1.0 m$^3$ h$^{-1}$ and 170 g L$^{-1}$, 1.5 m$^3$ h$^{-1}$ and 145 g L$^{-1}$, and 1.3 m$^3$ h$^{-1}$ and 120 g L$^{-1}$. Obviously, the salt concentration of the drainage water in the SPD test area gradually decreased as the leaching times (DL1 to DL3) increased. That is, the salt concentration of the drainage water decreased 5.3 g L$^{-1}$ for every 1% soil desalination rate decrease in 0–40 cm soil layers. In addition, we found that the average drainage flow of the leaching (DL2) test in September was 0.3 and 0.5 m$^3$ h$^{-1}$ higher than April (DL3) and June (DL1), respectively. The maximum drainage flow of DL2 was 1.0 and 1.2 m$^3$ h$^{-1}$ higher than DL3 and DL1, respectively, and did not decrease with lower soil desalination rate; thus, we

considered that the optimum time to improve saline–alkali soil by SPD drainage measures was September in each year, which could replace farmland irrigation in winter.

The average soil desalination rates were 2.8% (DL1), 5.3% (DL2), and 6.9% (DL3), and the average drainage flow and salt concentration of the drainage water were 3.9 m$^3$ h$^{-1}$ and 120 g L$^{-1}$, 3.3 m$^3$ h$^{-1}$ and 130 g L$^{-1}$, and 4.9 m$^3$ h$^{-1}$ and 140 g L$^{-1}$, respectively. Obviously, the average soil desalination rates and salt concentration of the drainage water in the ODD test area gradually increased as the leaching times increased. That is, the salt concentration of drainage water increased 4.9 g L$^{-1}$ for every 1% soil desalination rate decrease in 0–40 cm soil layers. In addition, the change trend of drainage flow in the ODD test area had a strong linear correlation with the month; the drainage flow of the leaching test in April (DL3) was 0.9 and 1.2 m$^3$ h$^{-1}$ higher than June (DL1) and September (DL2), respectively. However, we did not consider that the optimum time to improve saline–alkali soil by ODD drainage measures was April in each year because the single measure (ODD) cannot effectively decrease the soil EC in the root layers and thus cannot solve the problem of soil salinization within one to two years.

In general, the average soil desalination rate in the SPD test area was one order of magnitude higher than ODD. In arid areas, drip irrigation and leaching, combined with SPD drainage measures, had a highly significant effect when used to improve saline–alkali soil, with the soil EC decreasing to less than 5 dS m$^{-1}$. This implies that the soil desalination rate in the SPD test area was decreased from moderately saline soil level (8−15 dS m$^{-1}$) to weakly saline soil level (4−8 dS m$^{-1}$) [31]. Although the current results achieved the phased target of improving saline–alkali cultivated land, the soil EC did not decrease to the slightly saline soil level (2–4 dS m$^{-1}$) or non−saline soil level (<2 dS m$^{-1}$). Thus, further work is needed to assess the long-term impact of SPD and ODD drainage measures (e.g., effect of salt stress on root growth; secondary salinization).

*4.3. Effect of Soil EC on Cotton Growth in Saline–Alkali Cultivated Land*

In arid areas, soil salinization is the main factor inhibiting cotton growth [32,33]. Zhang et al. [34] showed that cotton seeds could germinate normally in soil with electrical conductivity less than 5 ds m$^{-1}$ and could still grow normally in weakly saline soils (4–8 ds m$^{-1}$). This result was consistent with Akhtar et al. [35], who showed that the critical value of cotton seed germination was 8 ds m$^{-1}$. Dong et al. [36] confirmed that when the soil EC is greater than 7 dS m$^{-1}$, the yield and quality of cotton is suppressed, and the dry weight per plant of cotton decreases by 75%. Wang et al. [37] concluded that the critical value of soil EC affecting cotton is 8 dS m$^{-1}$ during seed germination, which directly affects the plant height. The above results are consistent with the present study. The initial soil EC in 2015 was 15.5–16.1 dS m$^{-1}$, which inhibited the normal germination of cotton. After the first salinity leaching test in June 2016, the soil EC in the 0–40 cm soil layer was 10.4–10.9 dS m$^{-1}$, which was still not suitable for cotton planting. Therefore, oil sunflower with stronger salt tolerance was planted, but the germination rate was maintained below 50%, which had no economic value. After the third salinity leaching test in April 2017, the soil EC in the 0–40 cm soil layer was 3.4–5.1 dS m$^{-1}$. We set out to try to grow cotton in SPD test areas, and the seed cotton yields of P1, P2, P3 were 3.3, 3.8, and 3.8 t ha$^{-1}$, respectively in 2017. It can be said that SPD was the direct cause of the decrease of soil EC and played a key role in the increase of seed cotton yield. This study found that the mulch with drip irrigation and subsurface pipe drainage (MDI-SPD) could increase seed cotton yield by 27% in salinized farmland with soil EC between 3.4 and 5.1 dS m$^{-1}$. Qayyum and Malik [38] suggested that the seed cotton yield in slightly saline soil (3–5 dS m$^{-1}$) was 41% lower than that non-saline soil (0–3 dS m$^{-1}$). This means that if the MDI-SPD system is further adopted to decrease the soil EC from the current 5.1–7.6 dS m$^{-1}$ to below 3 dS m$^{-1}$, there is still a potential for substantial increase in seed cotton yield. Another notable finding was the effect of salt carried by groundwater up to the tillage layers (0–40 cm) might be substantially decreased when the groundwater level declines from 1–2 m to 4–5 m.

Therefore, the seed cotton yield differences in the SPD and ODD test areas observed in the present study were mainly attributable to the groundwater level changes.

In summary, salinity leaching and subsurface pipe drainage are effective remedial measures for sustainable agriculture on saline–alkali cultivated lands. Under this management method, the land proportion affected by salinization reduced by 17%, and the planting intensity of crops increased by 55% [39]. The testing of subsurface drainage systems carried out by Ritzema et al. [40] in India showed that the soil salt content can be reduced by 25–50% during one growth period, and the yield of rice, cotton, sugarcane, and wheat can be increased by 69%, 64%, 54%, and 136%, respectively. Further research should be undertaken to explore how to enhance prevention and control soil salinization during snowmelt and flood seasons, how to use SPD with drip irrigation and leaching to save water, how to resolve the problem of deep seepage of salt, and how to resolve the secondary salinization caused by excessively high groundwater mineralization.

## 5. Conclusions

An experiment was carried out in 2016–2017 in two fields on the edge of the Gurbantunggut Desert. The purpose of the current study was to determine the improvement effect on saline–alkali soil with SPD and ODD drainage measures, including soil EC, groundwater, and crop growth. The study disclosed that the soil EC in the SPD test area showed a stepped decrease over time, with the average annual decrease reaching 10 dS m$^{-1}$ in 0–40 cm soil depths and the soil desalination rate ranging from 25.8–35.2%. In the ODD test area, the further away from the open ditch, the more the soil EC decreased; however, the soil EC remained between 9–12 dS m$^{-1}$, and the soil desalination efficiency of ODD was only 1/10 of SPD. The average seed cotton yield of the SPD and ODD test areas was 3.6 and 2.4 t ha$^{-1}$, respectively, The closer the horizontal distance to the subsurface pipe or open ditch, the lower the oil sunflower emergence percentage, plant height, dry matter mass, seed cotton yield, and cotton dry matter mass. A high level of soil EC (9–12 dS m$^{-1}$) and groundwater level (3 m) still remained by far the most dominating limiting factors for plant growth in the ODD test area.

Additionally, our research showed that the critical depth of groundwater level affecting soil secondary salinization of saline−alkali cultivated land was 4 m. The drainage flow of SPD and ODD test areas exhibited a slight change from 2.0 to 3.2 m$^3$ h$^{-1}$ and 3.0 to 5.0 m$^3$ h$^{-1}$, respectively, and the salt concentration of groundwater was 40 to 110 g L$^{-1}$ and 46 to 70 g L$^{-1}$, respectively. Further research should focus on working with the current irrigation systems to achieve the goal of sustainable agriculture development.

**Author Contributions:** Conceptualization, L.Z.; methodology, L.Z.; software, L.Z.; formal analysis, L.Z.; data curation, L.Z.; writing—original draft, L.Z.; visualization, L.Z.; conceptualization, T.H.; supervision, T.H.; project administration, T.H.; funding acquisition, T.H.; data curation, L.Y.; methodology, X.X.; validation, X.X.; data curation, X.X.; writing—original draft, X.X.; investigation, resources, Y.F.; supervision, Y.F. All authors have read and agreed to the published version of the manuscript.

**Funding:** This research was funded by the National Natural Science Foundation of China [grant number 51969027, U1803244], Major Scientific and Technological Projects of XPCC in China (2017AA002, 2021AB021), China Scholarship Council (201909505014) and Shihezi University (grant numbers CXRC201801, RCZK2018C22).

**Institutional Review Board Statement:** Not applicable.

**Informed Consent Statement:** Not applicable.

**Data Availability Statement:** Not applicable.

**Acknowledgments:** The authors are grateful to the anonymous reviewers and the corresponding editor for their helpful and constructive comments and suggestions for improving the manuscript.

**Conflicts of Interest:** The authors declare no conflict of interest.

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
