# Peer review of "Study on the Farmland Improvement Effect of Drainage Measures under Film Mulch with Drip Irrigation in Saline–Alkali Land in Arid Areas"

_sustainability, doi:10.3390/su13084159_

Round 1

Reviewer 1 Report

The manuscript is a field study regarding the effect of 2 types of drainage in combination with drip irrigation in order to both desalinate soil and to allow for crop cultivation. The study is of high interest to allow for crops production in arid/saline environments, pretty well described and well organized. However I have the following concerns regarding the manuscript that have to be addressed before the manuscript been accepted for publication.

  • I miss a general description of the concept (schematic or sthing) regarding water that should describe both drip irrigation system, drainage systems, ground water level in order to better understand the overall concept
  • The authors should provide a better description and analysis of the water that was used both for drainage and drip irrigation.
  • My understanding is that crops were obtained during the experiments when the salinity was constantly decrease (soil salinity) and as such it is a bit of a transition phase. It would have been great to compare crop growth vs. initial statement (i.e. crop growth in saline environment or some kind of baseline experiments)
  • Figure 5: my understanding is that salt leaching from the soil leads to increased salinity of the groundwater. How would that be in a longer term/more intensive treatment? Could the groundwater salinity be stabilized or including could decrease? Drainage and irrigation will lead to groundwater level increase or not?

Author Response

Point 1: I miss a general description of the concept (schematic or sthing) regarding water that should describe both drip irrigation system, drainage systems, ground water level in order to better understand the overall concept

Response 1: I have added in the introduction. drip irrigation system and ground water level: In the past two decades, drip irrigation system, which can increase soil moisture and prevent soil secondary salinisation, has been widely implemented in cotton cultivation in Xinjiang, to combat drought and water shortages. However, drip irrigation system can only regulate the soil salinity in the root layer of crops, which cannot fundamentally decrease the soil salinity. In addition, drip irrigation system has a certain uplifting effect on the groundwater level, and has a potential threat to the growth of crops in areas with higher groundwater levels. Drainage systems: Drainage system can effectively solve the problem of soil salinization, and control the groundwater level, such as subsurface pipes, open ditch drainage systems.

Point 2: The authors should provide a better description and analysis of the water that was used both for drainage and drip irrigation.

 Response 2: I have added in the introduction. Drainage and drip irrigation: Drainage system combined with drip irrigation can directly and effectively decrease soil salinity and groundwater level in unsaturated root zone, thus achieving the purpose of improving saline cultivated land.

Point 3: My understanding is that crops were obtained during the experiments when the salinity was constantly decrease (soil salinity) and as such it is a bit of a transition phase. It would have been great to compare crop growth vs. initial statement (i. e, crop growth in saline environment or some kind of baseline experiments)

Response 3: I have added Figure 7 and its result. Oil sunflower and cotton growths were in a transition stage when the soil EC was constantly decrease (Figure 6). In 2016, The highest correlations were seen between the changes of soil EC and oil sunflower plant height, followed by emergence percentage (R2=0.64) and dry matter mass (R2=0.63). Likewise, in 2017, The highest correlations were seen between the changes of soil EC and cotton plant height (R2=0.86), followed by emergence percentage (R2=0.83), dry matter mass (R2=0.76), and seed cotton yield (R2=0.74). Overall, the oil sunflower plant height, emergence percentage, and dry matter mass status increased significantly when the soil EC change decreased from 9 to 5 dS m-1; the cotton plant height, emergence percentage, dry matter mass, and seed cotton yield status increased significantly when the soil EC change decreased from 4 to 1 dS m-1.

Figure 7. Effects of soil EC decrease on growth of oil sunflower (a) and cotton (b).  represents the threshold of soil EC.

Point 4: Figure 5: my understanding is that salt leaching from the soil leads to increased salinity of the groundwater. How would that be in a longer term/more intensive treatment? Could the groundwater salinity be stabilized or including could decrease? Drainage and irrigation will lead to groundwater level increase or not?

Response 4: I have added in Discussion 4.1. In 1−4 months after leaching and SPD measures, we found that the salinity of groundwater continued to increase, and the groundwater level was also rising gradually, but the upward trend lasted for up to 4 months, and then began to gradually decrease (Figure 6). When leaching and SPD measures decrease the soil EC to a slightly saline level (1−3 dS m-1), we will reduce the frequency of using SPD. (According to our latest SPD test results in 2020, the groundwater level has been declined to 8 m, the groundwater salinity has decreased by 30%, and the cotton yield has reached 6.9 t ha-1).

Reviewer 2 Report

My only comment is about normality test (lines 319 - 320).

Kolmogorov Smimov test results could be included.

Alternatively Q-Q plots to explain normality could be shown.

Author Response

Point 1: My only comment is about normality test (lines 319 - 320). Kolmogorov Smimov test results could be included.

Response 1: Application of Kolmogorov−Smimov test show that the baseline data are non−homogeneous (P=0.004), the soil desalination rate of SPD and ODD test areas are not draw from the same distribution, and thus the differences between these distributions are statistically significant (p < 0.05). Alternatively Q-Q plots shows that the soil desalination rate of each treatment did not follow the normal distribution (Figure 4).

Point 2: Alternatively Q-Q plots to explain normality could be shown.

Response 2: I have added Figure 4.

Figure 4. Normal Q-Q plot of soil desalination rate

Round 2

Reviewer 1 Report

The mansucript has been improved with regards to my comments.

I still have a comment. In fig.7, it shows that production increase with increased soil conductivity...which is different from the argument discussed. Please clarify this point or correct the fig if this is the issue.

Author Response

Response 1: I'm sorry that I didn't modify this problem in place..which made you confused. In fig.7, by this I mean the independent variable "the extent of soil EC decrease", from 1.0 to 4.5 dS cm-1. The greater the decrease value of soil EC, the higher the seed cotton yield. Among them, the “△ soil EC” on the horizontal axis is not commonly used, I would like to try to replace “△ soil EC” with “the decrease value of soil EC”, is this ok with you?

Round 3

Reviewer 1 Report

Ok. Sorry I did not see the delta...but I guess that was a bit confusing anyway for the potential readers. I would have suggested to put the actual values of onductivity in x axis but this way it is ok.